

# Oxidative stress as a potential target in acute kidney injury

Anamaria Magdalena Tomsa[1,*], Alexandru Leonard Alexa[2], Monica Lia Junie[3,*], Andreea Liana Rachisan[1] and Lorena Ciumarnean[4]

[1] Department of Pediatrics II, University of Medicine and Pharmacy of Cluj-Napoca, Cluj-Napoca, Romania
[2] Department of Anesthesia and Intensive Care I, University of Medicine and Pharmacy of Cluj-Napoca, Cluj-Napoca, Romania
[3] Department of Microbiology, University of Medicine and Pharmacy of Cluj-Napoca, Cluj-Napoca, Romania
[4] Department of Internal Medicine IV, University of Medicine and Pharmacy of Cluj-Napoca, Cluj-Napoca, Romania
* These authors contributed equally to this work.

Corresponding author
Andreea Liana Rachisan,
andreea_rachisan@yahoo.com

## ABSTRACT

**Background:** Acute kidney injury (AKI) is a major problem for health systems being directly related to short and long-term morbidity and mortality. In the last years, the incidence of AKI has been increasing. AKI and chronic kidney disease (CKD) are closely interconnected, with a growing rate of CKD linked to repeated and severe episodes of AKI. AKI and CKD can occur also secondary to imbalanced oxidative stress (OS) reactions, inflammation, and apoptosis. The kidney is particularly sensitive to OS. OS is known as a crucial pathogenetic factor in cellular damage, with a direct role in initiation, development, and progression of AKI. The aim of this review is to focus on the pathogenetic role of OS in AKI in order to gain a better understanding. We exposed the potential relationships between OS and the perturbation of renal function and we also presented the redox-dependent factors that can contribute to early kidney injury. In the last decades, promising advances have been made in understanding the pathophysiology of AKI and its consequences, but more studies are needed in order to develop new therapies that can address OS and oxidative damage in early stages of AKI.

**Methods:** We searched PubMed for relevant articles published up to May 2019. In this review we incorporated data from different types of studies, including observational and experimental, both in vivo and in vitro, studies that provided information about OS in the pathophysiology of AKI.

**Results:** The results show that OS plays a major key role in the initiation and development of AKI, providing the chance to find new targets that can be therapeutically addressed.

**Discussion:** Acute kidney injury represents a major health issue that is still not fully understood. Research in this area still provides new useful data that can help obtain a better management of the patient. OS represents a major focus point in many studies, and a better understanding of its implications in AKI might offer the chance to fight new therapeutic strategies.

## INTRODUCTION

Acute kidney injury (AKI) represents a syndrome in which the renal function deteriorates due to a sudden drop in the glomerular filtration rate. Traditionally, it has been viewed in an anatomical context (prerenal, intrarenal, and postrenal), while at the same time it has been considered to rarely transition to chronic kidney disease (CKD) (*Chawla et al., 2014*). Nowadays, the emphasis has been put on an improved understanding of AKI and its pathophysiological mechanisms, as it is considered a major risk factor for developing CKD. The fact that each episode of AKI significantly influences the outcome and accelerates the development of CKD, while a patient who suffers from CKD is more predisposed to suffer new episodes of AKI, has led to the idea that these entities, previously considered independent from each other, actually represent an interconnected syndrome (*Chawla et al., 2014*; *He et al., 2017*).

Up until recently, the real incidence of AKI was not known due to the fact that there was no consensus on what AKI actually represents. The lack of a standard definition for AKI led to different approaches to these patients, with each medical center using its own definition for AKI (*Mehta & Chertow, 2003*). In 2004 the first standard definition was created, based on the RIFLE criteria (Risk, Injury, Failure, Loss, End-stage kidney disease) by the Second International Consensus Conference of the Acute Dialysis Quality Initiative (ADQI) Group (*Bellomo et al., 2004*). Nowadays, the reported incidence shows that this syndrome represents a major public health problem, with about one in four hospitalized patients worldwide (*Palevsky et al., 2013*). The mortality rates vary by age, with 23.9% in adults and 13.8% in pediatric patients (*Susantitaphong et al., 2013*). The severity of this syndrome in the affected patients is considered to be an independent risk factor for their outcome and mortality (*Ostermann & Chang, 2007*; *Bagshaw et al., 2008*).

Emerging evidence highlights that AKI represents a major risk factor for developing CKD independent from the renal function recovery in the affected patients, even in those who seemed to have completely recover after an episode of AKI (*Lameire et al., 2013*; *Wald et al., 2009*; *Basile et al., 2016*). Recent experimental results showed that the transition from AKI to CKD may be caused by a maladaptive cellular response or by a reparative response that has been misdirected or maladaptive (*Basile et al., 2016*; *Agarwal et al., 2016*; *Takaori et al., 2016*). Also, oxidative stress (OS) plays a crucial role in both AKI and CKD, as it has been considered a central aggravating factor (*Tanaka, Tanaka & Nagaku, 2014*; *Ruiz et al., 2013*). Excessive accumulation of reactive oxygen species (ROS) determines the activation of adaptive gene programs, which experimental mouse models suggest they offer insufficient protection (*Nezu et al., 2017*).

This review aims to summarize the current knowledge about the role that OS plays in the pathophysiology of AKI, in order to gain a better understanding of this ubiquitous pathology. New biomarkers and new therapeutic strategies are needed in order to address this syndrome; therefore, a detailed knowledge about its underlying mechanisms is

mandatory. At this moment, there is little data about this subject in the literature, even though OS represents a major focus point in many diseases. In this review we detailed the main sources of OS in AKI, their mechanisms of action and their effect on different cellular structures. Furthermore, we included the main antioxidants that can be found in the renal tissues and data about OS biomarkers. We address this review to nephrology residents and specialists, and to whomever wants to gain a more in-depth view in this particular subject.

## SURVEY METHODOLOGY

We completed an electronic literature search in the PubMed database and we included pertinent articles published up to May 2019, that provided information about OS and its role in the pathophysiology of AKI. In our search we used the following terms: "AKI," "acute kidney failure," "acute kidney insufficiency" in combination with "OS," "ROS" or "reactive nitrogen species (RNS)." In order to write this review, we comprised data from numerous types of studies, including observational and experimental studies, both in vitro and in vivo, including randomized controlled ones. For the purpose of this review we used overall selected papers.

### Oxidative stress overview

Oxidative stress reprexidants, ROS and/or RNS which surpasses the endogenous antioxidant capacity, due to various causes. Defined for the first time in 1985 by Stahland and Sies, OS has recently been intensely studied as a regulatory element in various diseases. A free radical (FR) is defined as a species that presents a single or more unpaired electrons, and which exists autonomously (*Halliwell, 2006*). FRs are considered essential in biological evolution as they play a major role in several biochemical reactions (*Singh et al., 2019*).

Up to this moment, OS has been proved to represent a pathogenetic factor in cardiovascular diseases, neurodegenerative diseases, cancer, aging, and many others (*Valko et al., 2007*; *Singh et al., 2019*). OS also represents a major factor in the development of kidney damage (*Himmelfarb et al., 2004*), therefore it might play a crucial role in therapeutic intervention when targeted. Also, OS presents damaging effects on biomolecules such as DNA, RNA, proteins, lipids, enzymes, etc. The changes that reactive species (RS) make upon these molecules could be highlighted and used as biomarkers for OS (*Singh et al., 2019*).

### Are reactive species important in maintaining homeostasis?

It is considered that RS are important for maintaining homeostasis when they are present in low concentrations, in certain compartments, at a certain moment. ROS and RNS are responsible for normal redox signaling, and therefore they represent a promotor for cell survival, growth, and proliferation (*Ratliff et al., 2016*).

In the kidney, the blood flow and the glomerular filtration rate are maintained at a constant level by renal autoregulation, despite all the physiological changes or pathological states that may take place in the body. Two key mechanisms are responsible for the renal autoregulation: the myogenic response and tubuloglomerular feedback. Studies show

that both vasoreactivity and the myogenic response are influenced by the presence of RS such as nitric oxide (NO) and superoxide ($O_2^-$) (*Carlström, Wilcox & Arendshorst, 2015*; *Just, 1997*; *Loutzenhiser et al., 2006*; *Navar et al., 2008*).

Nitric oxide is a vasodilator factor generated by endothelial NO synthase (eNOS), and it defends the renal cells (endothelial and mesangial cells) from fibrosis and apoptosis by inducing the expression of antioxidative genes and promoting a physiological renal hemodynamics (*Ratliff et al., 2016*). It is well known that NO is a major factor in modulating the myogenic response (*Carlström, Wilcox & Arendshorst, 2015*), with numerous studies showing that inhibition of NOS decreases renal blood flow and increases the renal vascular resistance in both pressor dosages (*Baylis & Qiu, 1996*; *Majid, Williams & Navar, 1993*) as well as in dosages that have no impact on the blood pressure (*Carlström, Wilcox & Arendshorst, 2015*; *Deng & Baylis, 1993*). Also, the infusion of NO donors (sodium nitroprusside) and NO precursors (L-arginine) seems to counteract renal vasoconstriction (*Kiyomoto et al., 1992*; *Kumagai et al., 1994*).

Intracellularly, low levels of NO may inhibit cytochrome-c oxidase, altering the generation of ROS in the mitochondria (*Brown, 1995*). When ROS produced by mitochondria increase moderately, they stabilize hypoxia-inducible factor (HIF) in endothelial cells, and stimulate the nuclear factor erythroid 2–related factor 2 (Nrf2) (*Piantadosi & Suliman, 2006*; *Guzy et al., 2005*). Both HIF and Nrf2 protect the renal tissues against OS, therefore ROS may exhibit renoprotective effects in certain concentrations (*Ratliff et al., 2016*; *Nangaku & Eckardt, 2007*). Nrf2 represents the main regulator of the response to OS (*Ruiz et al., 2013*).

Interestingly, one study that focused on the effects of endogenous $O_2^-$ showed that it becomes a crucial factor in maintaining a physiological tone in the renal vasculature when it is induced by vascular NADPH oxidase (NOX) (*Haque & Majid, 2004*). This further demonstrates that RS might play a crucial role in maintaining homeostasis.

### What reactive species are generated in AKI?

The main etiologies of AKI are linked to ischemia and hypoxia. Due to the decrease in the renal blood flow, the cellular nutrient and oxygen uptake is limited, leading do the development of inflammation and acute tubular necrosis (*Basile, Anderson & Sutton, 2012*). Renal ischemia and reperfusion represent major triggers of ROS with consequential damage in renal function and tissue integrity (*Deng & Baylis, 1993*). In sepsis-induced AKI there is an extensive immune response that is responsible for renal vasoconstriction, endothelial injury, and localized hypoxia, which ultimately triggers the formation of ROS (*Andrades et al., 2011*). Further, uremia is highly associated with an increase in circulating levels of indole and carbonyl compounds, which are able to upregulate systemic OS (*Tanaka et al., 2017*).

Some of the initial changes that are observed with AKI development are depletion of ATP and changes in the structure of the mitochondria, leading to alterations and dysfunction in the energetic metabolism (*Kiyomoto et al., 1992*; *Tanaka et al., 2017*; *Tracz et al., 2007*). Each type of renal cell presents a different number of mitochondria, which is considered to be one of the major sources of ROS (*Duchen & Szabadkai, 2010*),

therefore the production of ROS varies between the renal structures. Mitochondria is also responsible for producing molecules that counteract the OS (*Singh et al., 2019*). In the kidney, NOX and the mitochondrial respiratory chain are considered to be the main sources of ROS production (*Sureshbabu, Ryter & Choi, 2015*).

One study evaluated the mitochondrial function, structure and redox state in rodents with induced nephrotoxic and ischemic AKI via in vivo exogenous and endogenous multi-photon imaging. The results showed changes in mitochondrial NADH levels and proton motive force, and upregulated levels of mitochondrial $O_2$, along with disjointed mitochondria. They concluded that mitochondria represents a major source of ROS, and that an alteration in mitochondrial function plays an important role in the early phase of renal ischemia (*Hall et al., 2013*; *Tanaka et al., 2017*). In contrast, after gentamycin exposure, the first alterations to appear were at lysosomal level in the renal epithelium, along with anomalies in brush border cells. The mitochondrial dysfunction, with changes in morphology of the mitochondria, alterations in NADH levels, RS and altered proton motive force occurred later in the development of nephrotoxic AKI (*Tanaka et al., 2017*; *Deisseroth & Dounce, 1970*). Mitochondria is, therefore, directly responsible for increased levels of OS in AKI.

Despite the abundance of studies conducted on this topic, the exact mechanism through which RS are generated in AKI remains unknown. However, OS might represent a potential therapeutic target. Table 1 summarizes the main ROS found in AKI.

The production of the *superoxide anion $O_2^-$* is the result of the one-electron reduction of oxygen in its molecular form (*Noiri, Addabbo & Goligorsky, 2011*). Superoxide can be generated by a large variety of oxidase enzymes, and can also be generated inside the mitochondria by components of the electron transport chain. $O_2^-$ is mainly transported through anion channels, therefore the diffusion across different membranes is limited. Superoxide anion is a rather selective FR, which is able to form a non-radical ROS ($H_2O_2$) via dismutation. The reaction can take place spontaneously, but it can also be facilitated by enzymatic catalysis (*Ratliff et al., 2016*). The presence of the superoxide anion triggers a cascade of events, as its presence leads to the generation of other ROS. $O_2^-$ may also be produced from xanthine by xanthine oxidase (XO), and from NADPH and NADH by various oxidase enzymes which are induced by an inflammatory response (*Deng & Baylis, 1993*). The most potent action of $O_2^-$ is represented by the scavenging of NO. As the levels of superoxide anion increases, it is able to disrupt the iron-sulfur centers, and it may react with catecholamines (*Ratliff et al., 2016*). Local ischemia and cytokines generated in AKI induced by sepsis activate the endothelium of the renal vasculature and recruit cells from the immune system that are able to generate $O_2^-$ via NOX (*Kiyomoto et al., 1992*).

*Hydrogen peroxide ($H_2O_2$)* can be generated by dismutation but also by oxidases which are able to directly reduce the molecular oxygen. $H_2O_2$ diffuses across different biological membranes in a similar way to water, which makes it able to express its oxidative properties in other cellular compartments and even in other cells. Peroxides are able to react with different molecules containing iron, leading to generation of additional ROS.

Table 1 The main reactive species in AKI.

| | | | | | |
|---|---|---|---|---|---|
| Superoxide anion (Ratliff et al., 2016; Deng & Baylis, 1993; Kiyomoto et al., 1992; Basile, Anderson & Sutton, 2012) | $O_2^-$ | Induced by NOX, XO, and other enzymes Generated by the mitochondrial respiratory chain | Maintains a physiological role in the renal vasculature | Generates other ROS (e.g., forms $H_2O_2$ via dismutation) Scavenges NO Disrupts the iron-sulfur centers | In sepsis-induced AKI, $O_2^-$ is generated by immune cells |
| Hydrogen peroxide (Ratliff et al., 2016; Dennis & Witting, 2017) | $H_2O_2$ | Generated by dismutation Generated by oxidases from molecular oxygen | | Diffuses across biological membranes to other cellular compartments and other cells Reacts with iron-containing molecules, releasing additional ROS | |
| Hydroxyl radical (Brown, 1995; Bogdan, 2001; Dennis & Witting, 2017) | $HO^-$ | Generated by Fenton reaction | | Lipid peroxidation, with subsequent membrane damage Generates additional ROS | |
| Hypochlorous acid (Ratliff et al., 2016; Kiyomoto et al., 1992) | HClO | Generated by MPO in inflammatory cells | | Lipid soluble molecule Reacts with amines, producing chloroamines | |
| Peroxynitrite (Ratliff et al., 2016) | $ONOO^-$ | Generated from nitric oxide reacting with superoxide anion Generated by heme peroxidase enzymes from the nitrate metabolism | Major inhibitor of the mitochondrial respiration chain | Oxidizes or nitrates thiols Oxidizes and nitrates fatty acid Irreversibly disrupts the centers of iron-sulfur | |
| Nitric oxide (Ratliff et al., 2016; Carlström, Wilcox & Arendshorst, 2015; Kiyomoto et al., 1992; Kumagai et al., 1994; Brown, 1995; Ling et al., 1999) | NO | Generated by eNOS and eNOS | Vasodilator and counteracts renal vasoconstriction Modulates the myogenic response Protects endothelial and mesangial cells from fibrosis and apoptosis Induces antioxidative genes Inhibits cyt-c oxidase and alters the generation of ROS in the mitochondria Binds to guanylate cyclase and regulates the production of cGMP | Soluble gas Larger levels of NO interact with bound iron, and produce NO-derived RNS that can nitrosate thiols | Mice deficient in iNOS were resistant to renal IRI |

When peroxides react with $Fe^{2+}$ they generate the hydroxyl radical ($HO^-$), known as the Fenton reaction. When peroxide is metabolized by specific heme peroxidase in the presence of other molecules such as chloride or nitrite, it can generate hypochlorous acid, and nitrogen dioxide (Ratliff et al., 2016; Dennis & Witting, 2017).

*The hydroxyl radical ($HO^-$)* may be generated by the Fenton reaction. This radical alters and reacts with almost every cellular component, generating additional ROS

(*Brown, 1995*). It produces lipid peroxidation with subsequent membrane damage and toxic compounds release, including aldehydes (*Dennis & Witting, 2017*).

*Hypochlorous acid (HClO)*, a highly reactive lipid soluble molecule (*Ratliff et al., 2016*), is generated by phagocyte myeloperoxidase in inflammatory cells when local ischemia is present (*Kiyomoto et al., 1992*), and it may react with amines, producing chloroamines.

*Peroxynitrite (ONOO$^-$)* is generated when superoxide anion reacts with NO. The spontaneous decomposition of peroxynitrite generates nitrogen dioxide (NO$_2$). Also, NO$_2$ is generated by heme peroxidase enzymes from the nitrate metabolism. These RS are able to activate a variety of signaling mechanisms by oxidizing or nitrating thiols, sometimes influencing processes that are also targeted by peroxide. When RNS react with unsaturated fatty acids (FA), the result is oxidized and nitrated FA which present a multitude of biological actions. ONOO$^-$ seems to be a major inhibitor of the mitochondrial respiration chain, irreversibly disrupting the centers of iron-sulfur (*Ratliff et al., 2016*).

*Nitric oxideNO* represents a soluble gas that is able to bind to ferrous sites of heme. It is generated by NOS enzymes, and in normal amounts it is a part of physiological signaling processes: it binds to guanylate cyclase in order to regulate the production of cGMP, and it binds to cytochrome c oxidase to regulate the mitochondrial respiration. While it is mainly generated by regulating the activity of nNOS (NOS1) and eNOS (NOS3), an inflammatory status is able to induce iNOS (NOS2) expression (iNOS—inducible NO synthase; nNOS—neuronal NO synthase). This process leads to the generation of larger quantities of NO which consequently modulates the inflammatory processes. Larger levels of NO interact with bound iron, and produce NO-derived RNS that can nitrosate thiols. When the concentration of NO equalizes the levels of local antioxidant enzymes, it competes with them for the scavenging of O$_2^-$, generating ONOO$^-$(*Ratliff et al., 2016*; *Dennis & Witting, 2017*). However, it has been shown that when using agents to inhibit the global production of NO, including the NO production from constitutive eNOS, the effect on renal ischemia-reperfusion injury (IRI) was not renoprotective (*Yaqoob, Edelstein & Schrier, 1996*).

Another study showed that mice who were deficient in iNOS were resistant to renal IRI, concluding that a constant release of NO from NOS2 could be a pathogenetic factor (*Ling et al., 1999*). NOS2 is constitutively expressed in the renal tissue (*Thomas et al., 2015*), underlining that NO presents a physiological role in kidney functioning (*Araujo & Welch, 2006*). These mechanisms may potentially contribute to the development of renal disease.

Carbonyls, nitrated lipids, and unsaturated aldehydes are just some of the toxic reactive molecules that can be generated by ROS and RNS. These molecules are able to trigger signaling processes, at low levels of OS. As the level of OS increases, these molecules promote oxidative damage and cellular death (*Ratliff et al., 2016*; *Dennis & Witting, 2017*). Out of all possible etiologies for AKI, two of the most common are IRI and sepsis (*Piantadosi & Suliman, 2006*).

### Ischemia-reperfusion injury induced AKI

Ischemia-reperfusion injury represents a major cause of AKI, and the main cause of delayed renal graft function, and renal graft loss after kidney transplantation

(*Bogdan, 2001*). In IRI, the reperfusion phase is the crucial moment when the most IRI damage might occur. The initial event that takes place immediately after reperfusion is a sudden increase in superoxide anion production in the mitochondria which is released inside the cell, and represents the main trigger for the pathology that follows reperfusion (*Zweier, Flaherty & Weisfeldt, 1987*; *Chouchani et al., 2016*). In IRI, this mechanism plays a crucial role in initiating AKI, but also in maintaining it (*Nath & Norby, 2000*). Also, the early inflammatory response in IRI consists mainly of neutrophils, which generate ROS, and are recruited by ROS (*Friedewald & Rabb, 2004*).

One study that focused on the pathophysiology of renal IRI showed that pericytes express adhesion protein-1, a key protein that generates a local $H_2O_2$ gradient which further stimulates the neutrophil infiltration (*Tanaka et al., 2017*), proving that RS are directly involved in developing renal injury following ischemia.

Another study concluded that, following renal IRI, Nrf2-regulated cell defense genes presented significantly high levels in the kidneys of wild-type mice but not in Nrf2-knockout ($^{-/-}$) mice (*Leonard et al., 2006*; *Liu et al., 2009*). The same study showed that the loss of Nrf2 leads to an increase in severity of renal IRI, which also proves that OS plays a crucial role in the pathogenesis and outcome of renal IRI.

*Tracz et al. (2007)* and *Piantadosi & Suliman (2006)* showed that mice who were heme oxygenase-1 knockout (HO-1$^{-/-}$) had increased sensitivity to renal IRI, and expressed a higher inflammatory response and oxidative damage and an increase in mortality when compared to wild-type mice.

Another study demonstrated that mice deficient in transient receptor potential melastatin 2 (TRPM2) who were subjected to renal IRI showed resistance to OS and apoptosis (*Gao et al., 2014*). TRPM2 is a nonselective cation channel (for calcium, sodium, and potassium) activated by OS, ADP-ribose, tumor necrosis factor-$\alpha$ (TNF-$\alpha$), and intracellular calcium, all of which are considered to be increased in kidney ischemia. Gao et al. showed that TRPM2-knockout mice are resistant to IRI, with a reduction in OS, NOX activity, and apoptosis in the kidney. Also, they obtained similar results by pharmacologically inhibiting TRPM2. The cationic channel was identified mainly in the proximal tubule epithelial cells, and Gao et al. showed that its effects are attributable to expression in parenchymal cells. Also, they showed that the activation of RAC1, a constituent of the NOX complex, was promoted by TRPM2 following renal ischemia. In contrast, inhibition of RAC1 in vivo reduced ischemic injury and OS. This further shows that OS and ischemic injury are closely interrelated.

### Sepsis-induced AKI

In intensive care units (ICU), sepsis represents the major cause of AKI (*Uchino et al., 2005*). Of all the bacterial pathogens that induce AKI, Gram-negative bacteria represent a major threat due to the composition of their outer membrane. One of the main components of their outer membrane is represented by lipopolysaccharide (LPS), an endotoxin considered to be an important trigger of the inflammatory response in sepsis. These LPSs are recognized by Toll-like receptor (TLR), a transmembrane protein

expressed in renal tubules. Once the LPS bind to TLRs, the immune system is triggered, and the acquired immunity develops an antigen-specific response (*Medzhitov, 2001*).

TLR4 binds specifically to the LPS ligand, and their binding triggers a cascade of events that leads to OS and a pro-inflammatory response, with new cytokines and mediators being further released. TNF-α and interleukin-1β (IL-1β) are just two of the newly generated molecules which consequently promote the $H_2O_2$ formation, with oxidative damage which further enhances the inflammatory response (*Li & Engelhardt, 2006*).

Many studies suggest that both sepsis and ischemia upregulate TLR4 (*Wu et al., 2007*; *Pulskens et al., 2008*; *El-Achkar et al., 2006*). *Hato et al. (2015)* studied the effect of endotoxin preconditioning and concluded that it confers renal epithelial protection in vivo, by preventing peroxisomal damage, abolishing OS, and tubule injury. OS was measured in both preconditioned and nonpreconditioned mice, with measurements performed after 4 h from LPS inoculation intraperitoneally. High levels of OS were determined in proximal tubules of NP wild-type mice, whereas in preconditioned wild-type mice OS was absent. TLR4$^{-/-}$ mice showed no OS, which confirms that TLR4 plays a crucial role in LPS signaling pathway. Also, in preconditioned mice, KIM-1, and NGAL, markers for tubular injury, were significantly reduced, suggesting renal protection. This study proves that sepsis indeed upregulates TRL4, and leads to high levels of OS in renal proximal tubules, with subsequent oxidative damage.

TLR9 identifies bacterial DNA with unmethylated CpG motifs (*Tsuji et al., 2016*), but the TLR9 pathway is also activated by endogenous mitochondrial DNA (mtDNA) (*Zhang, Itagaki & Hauser, 2010*). Tsuji et al. (*Gao et al., 2014*) studied the role of mtDNA via TLR9 in septic AKI and showed that cytokine production, tubular mitochondrial disorders (ATP depletion), and splenic apoptosis are inflicted by the circulating mtDNA released in the early phase of sepsis. They used an animal model of sepsis in wild-type and TLR9$^{-/-}$ mice, but they also injected mitochondrial debris intravenously. The results show that TLR9$^{-/-}$ mice presented a lower bacteremia than wild-type mice, but with a higher level of leucocyte migration. Also, TLR9$^{-/-}$ mice presented lower levels of IFN-γ and IL-12 than wild-type mice, with attenuated production of superoxide in the proximal tubule cells. When injected with mitochondrial debris containing a substantial amount of mtDNA, wild-type mice had increased plasma levels of IL-12, but not of IFN-γ and TNF-α, while the levels of IL-12 in TLR9$^{-/-}$ mice did not increase as much. This study further shows that sepsis upregulates certain receptors that stimulate the production of RS and oxidative damage.

While sepsis remains ubiquitous, aforementioned studies suggest that the cascade of events that it triggers, including OS damage and inflammatory response, could be counteracted at receptor level. More studies are needed in order to gain a better perspective of how OS could be addressed in sepsis-induced AKI.

## How do the kidneys combat oxidative stress?

Homeostasis is favored by balancing the production of both oxidants and antioxidants. It has been proved that ROS play a physiological role in renal function, but when RS are unregulated or upregulated with consequent local accumulation, they are able to

induce OS, irreversibly damaging DNA, RNA, lipids, proteins, causing organelle dysfunctionality (*Dennis & Witting, 2017*). The oxidant production is usually counterbalanced by formation of endogenous antioxidants which disrupt the damaging effects of OS. High levels of ROS may overrun the antioxidant activity, promoting dysfunction of the vasculature, triggering an inflammatory response, with consequent cytotoxicity on renal tubule cells, mechanisms that are observed in the pathogenesis of AKI (*Ratliff et al., 2016*).

Antioxidants represent the body's system to combat the RS that are constantly released inside the cell (*Singh et al., 2019*). They are molecules that intervene early in AKI development, by scavenging RS. There are several antioxidant systems able to protect the renal cells from OS (*Ratliff et al., 2016*). Exogenous antioxidants are mainly represented by dietary and/or supplementation substances which downregulate OS, consequently diminishing lipid peroxidation, and oxidative damage (*Dennis & Witting, 2017*). Table 2 contains the main antioxidants that can be found in the renal tissue.

Modulating AKI with the help of antioxidants represents a focus point for many human and animal studies (*Chatterjee, 2007*), which have shown that nephrotoxicity and renal ischemia induce an increased oxidative damage with low levels of local antioxidants (*Paller, Hoidal & Ferris, 1984*; *Baliga et al., 1999*).

*Superoxide dismutase (SOD) isoforms* represent some of the major enzymes that fight against OS, and they all can be found in the kidneys. They are localized in both the extracellular space, and intracellular space (mitochondria, cytoplasm). SOD localization varies by species, but their renal activity is comparable in human, mouse, sheep, etc. (*Marklund, 1984*). SOD is a catalysator that helps the dismutation of $O_2^-$ into $O_2$ and $H_2O_2$. It is considered to be the first system to fight OS (*Ratliff et al., 2016*).

SOD1 (copper/zinc form of SOD) cannot be found inside the mitochondria, but is commonly present in other intracellular spaces. It is responsible for 80% of the activity that SOD has in the renal tissues (*Matés, 2000*), and for 1/3 of the SOD activity in the renal vasculature, where it stops disruption of NO signaling (*Carlström et al., 2010*). SOD1 suppresses the clearing of NO by $O_2^-$ inside the cellular compartments, maintaining the production of $H_2O_2$ (*Ratliff et al., 2016*).

SOD2 (manganese form of SOD) is present in great quantities inside the mitochondria, and it plays a key role in generating and releasing the peroxide from the mitochondria. $ONOO^-$ is able to inactivate SOD2. Overexpression of this isoform, but not catalase, ameliorates AKI induced by cisplatin in vitro (*Davis, Nick & Agarwal, 2001*), fact that shows the importance of superoxide anion in AKI (*Tanaka et al., 2017*).

One study in which SOD2 and SOD1 were ablated in mice, showed that the pathological phenotype was aggravated in SOD2-ablated mice, despite the fact that SOD1 is responsible for most of the SOD activity (*Schieber & Chandel, 2014*). This data suggests that the localization of RS is of great importance, indicating that mitochondria is crucial in the initiation and evolution of AKI (*Ratliff et al., 2016*).

SOD3 is an extracellular copper/zinc SOD isoform responsible for protecting NO from $O_2^-$, and also for converting $O_2^-$ into $H_2O_2$ in the extracellular environment (*Ratliff et al., 2016*).

**Table 2 The main antioxidants fighting oxidative stress in AKI.**

| | | | | |
|---|---|---|---|---|
| Superoxide dismutase | SOD (*Ratliff et al., 2016*; *Marklund, 1984*; *Matés, 2000*) | Extracellular and intracellular (mitochondria, cytoplasm) | Catalysator<br>Generates $O_2$ and $H_2O_2$ via dismutation<br>Considered the first system to fight oxidative stress | |
| | SOD1 (*Ratliff et al., 2016*; *Matés, 2000*; *Carlström et al., 2010*) (copper/ zinc isoform)<br>• dimer | Cytoplasm | Suppresses the clearing NO, maintaining the production of $H_2O_2$ | Gene located on chromosome 21 (21q22.1) |
| | SOD2 (*Tanaka et al., 2017*; *Davis, Nick & Agarwal, 2001*; *Schieber & Chandel, 2014*) (manganese isoform)<br>• tetramer | Mitochondria (great quantities) | Generates and releases peroxide from the mitochondria<br>Inactivated by $ONOO^-$ | Gene located on chromosome 6 (6q25.3)<br>Overexpression ameliorates cisplatin-induced AKI in vitro |
| | SOD3 (*Ratliff et al., 2016*) (copper/zinc isoform)<br>• tetramer | Extracellular | Protects NO from $O_2^-$<br>Converts $O_2^-$ into $H_2O_2$ | Gene located on chromosome 4 (4p15.3-p15.1) |
| Catalase (*Ratliff et al., 2016*; *Deisseroth & Dounce, 1970*; *Hwang et al., 2012*; *Vasko et al., 2013*) | Contains heme groups with iron core<br>• tetramer | Intracellular (mainly in in the cytosol and in peroxisomes)<br>In cells with aerobic metabolism | Converts $H_2O_2$ into $O_2$ and $H_2O$ | |
| Thioredoxin peroxidase systems (*Ratliff et al., 2016*) | | Intracellular | Converts $H_2O_2$ by oxidizing thioredoxin | |
| Glutathione and glutathione peroxidase family (*Araujo & Welch, 2006*; *Dennis & Witting, 2017*; *Oberley et al., 2001*; *Fukai, 2009*; *Doi et al., 2004*) | • tripeptide | Intracellular (cytoplasm, mitochondria, nucleus) and extracellular (pulmonary surfactant) | Reduce $H_2O_2$<br>Reduce lipid peroxides to lipid alcohols<br>Regulate apoptosis vs. necrosis<br>Modulate DNA synthesis and cellular division | |
| Edaravone (*Tanaka et al., 2017*; *Doi et al., 2004*; *Satoh et al., 2003*; *Aksoy et al., 2015*) | Norphenazone MCI-186<br>Synthetic medication | | Reduces ROS generation in the renal tubular cells, in vitro<br>Reduces lipid peroxidation, in vivo | Improves kidney function in rats with IRI, and in rats with nephrotoxicity |
| Vitamin C (*Ratliff et al., 2016*; *Tanaka et al., 2017*) | Vitamin<br>Dietary/ Supplementation intake | | Reacts with oxidized forms of enzymes and free radicals<br>Cofactor for some enzymatic reactions | Shown to improve kidney function in ischemia/ chemically/ rhabdomyolysis-induced AKI |
| Selenium (*Matés, 2000*; *Vasko et al., 2013*; *Oberley et al., 2001*; *Fukai, 2009*) | Trace element | | Participates in aerobic respiration reducing FR<br>Upregulates antioxidants | Deficiency linked to AKI<br>Improves kidney function in toxic AKI |
| Sulforaphane (*Hwang et al., 2012*; *Shokeir et al., 2015*) | Isothiocyanate<br>Dietary intake | | | Improves kidney function in renal IRI |

Superoxide dismutase mimetic activity can be found in pharmacologic agents Tempol and MnTMPyP which have proved to ameliorate both ischemia-induced AKI, and sepsis-induced AKI (*Chatterjee et al., 2000*; *Liang et al., 2009*). Also, MnTMPyp (which is both SOD and catalase mimetic) decreases SOD related to renal fibrosis after ischemic AKI (*Kim et al., 2009*).

*Catalase*, an enzyme containing heme, is responsible for further converting $H_2O_2$ into $O_2$ and $H_2O$, without requiring additional cofactors (*Ratliff et al., 2016*). It is ubiquitary in cells with aerobic metabolism, and high levels of catalase are found in renal cells (*Deisseroth & Dounce, 1970*). Intracellular disposition shows that catalase is mainly found in the cytosol and in peroxisomes (*Ratliff et al., 2016*). Its deficiency leads to ROS accumulation inside the mitochondrial matrix, causing mitochondrial dysfunction (*Hwang et al., 2012*). An experimental study of endotoxemia in mice showed that LPS can downregulate the catalase activity, aggravating renal damage (*Vasko et al., 2013*).

*Thioredoxin peroxidase systems* also consume $H_2O_2$, while oxidizing thioredoxin. The oxidized forms that result play a crucial role in several signaling mechanisms. These peroxidases can be found in most cellular compartments (*Ratliff et al., 2016*).

*Glutathione and glutathione peroxidase family* are present in all cellular compartments (including the nucleus, mitochondria, cytoplasm) (*Araujo & Welch, 2006*; *Dennis & Witting, 2017*; *Oberley et al., 2001*). Glutathione is important for regulating the cellular redox, while being a part of the controlling mechanisms responsible for the cellular proliferation, DNA synthesis, and cellular apoptosis (*Fukai, 2009*). In the mitochondria, glutathione regulates apoptosis and necrosis, while in the nucleus it modulates the cellular division (*Dennis & Witting, 2017*).

*Edaravone* represents an approved stroke therapy in Japan. It has been studied in multiple experimental models, and it has been shown that it reduces the generation of ROS in vitro in the renal tubular cells, reduces lipid peroxidation in vivo (*Tanaka et al., 2017*), improves kidney function in rats with IRI (*Doi et al., 2004*), and also in rats with nephrotoxicity (*Satoh et al., 2003*). These results show that edaravone (norphenazone, MCI-186) might be able to avoid preservation injury in kidney transplantation (*Tanaka et al., 2017*).

*Vitamin C* is a well-known antioxidant that attenuates OS, inflammation, and improves kidney function in AKI animal models (ischemia-induced, chemically-induced, rhabdomyolysis-induced AKI) (*Tanaka et al., 2017*). Vitamin C plays an important role for some enzymatic reactions as a cofactor, participating in several defense mechanisms. Vitamin C redox is preserved by diverse components of redox systems in the cell, reacting with oxidized forms of molecules (enzymes, FRs) (*Ratliff et al., 2016*).

*Selenium (Se)* represents a trace element that plays a role in cellular aerobic respiration by reducing FRs (*Aksoy et al., 2015*). Selenium deficiency was linked to AKI in rodent model (*Iglesias et al., 2013*). Also, several experimental studies showed improved kidney function after Se supplementation in AKI induced by cisplatin (*Matés, 2000*), and by gentamycin (*Randjelovic et al., 2012*). Selenium upregulated antioxidants in a pig model of transplanted kidneys (IRI) (*Třeška et al., 2002*).

*Sulforaphane*, a natural dietary isothiocyanate, has been shown to be renoprotective and able to improve kidney function in an animal renal IRI model (*Shokeir et al., 2015*). In this study, Shokeir et al. showed that ischemic preconditioning and sulforaphane both enhanced gene expression (Nrf2, heme oxygenase-1 HO-1, NADPH-quinone oxidoreductase1 NQO-1) and diminished the inflammatory (TNF-$\alpha$, IL-1, ICAM-1) and apoptotic markers (caspase-3), with better results for sulforaphane alone. Their combination improved the antioxidant gene expression and attenuated the inflammatory genes.

Antioxidants play a leading role in fighting OS and preventing oxidative damage. Both endogenous and exogenous (dietary/supplementation) seem to have a beneficial impact on renal function, with renoprotective effects and good outcomes even on the suffering kidneys.

### Are there any biomarkers for detecting oxidative stress in early AKI?

The interest in defining biomarkers for acute illnesses is continuously increasing. Recent studies focused on molecules that might increase specifically in OS. Some of them, conducted in patients who suffered kidney damage induced by sepsis or by other critically illnesses, highlighted the presence of plasmatic biomarkers from protein and lipid oxidation that could be correlated with other markers for cytokines, pro-oxidative mediators, and pro-inflammatory markers (*Himmelfarb et al., 2004*). Ischemia causes alterations in DNA structure and leads to lipids peroxidation, leading to increased levels of 3-nitrotyrosine, a well-known biomarker for ROS and RNS (*Noiri et al., 2001*; *Walker et al., 2001*).

One study showed that renal IRI causes increased urinary expression of thioredoxin1 (TRX), making it a possible biomarker for OS (*Piantadosi & Suliman, 2006*; *Kasuno et al., 2014*).

*Costa et al. (2018)* conducted a study in septic shock patients admitted to the ICU and determined the protein carbonyl concentration upon admission. The levels of protein carbonyl were significantly higher in patients who developed septic AKI, and they were also positively correlated with the SOFA score. Moreover, protein carbonyl concentration was associated with development of septic AKI, and with mortality in these critical patients. This study shows a novel molecule that could be used as a biomarker in AKI with excellent reliability.

Recently, transfer RNA (tRNA) has been shown to be cleaved into half molecules under OS, resulting tRNA-derived stress-induced fragments (tiRNAs) (*Yamasaki et al., 2009*). *Mishima et al. (2014)* used a specific tRNA-specific modified nucleoside 1-methyladenosine (m1A) antibody to show that changes in RNA structure occur much earlier than DNA damage in OS exposure. Change in tRNA structure leads to further tRNA fragmentation, reflecting early stages of OS damage. Authors suggest that detecting tRNA damage could be a valuable tool for recognizing early organ damage and making a better medical decision.

## CONCLUSIONS

Acute kidney injury, although common in clinical practice, is still poorly understood from the pathophysiological point of view. This leads to addressing AKI with only supportive therapies, instead of effective therapeutics to treat it. Research focuses on

understanding the molecular pathways that are present in renal injury, but the large possible etiologies of AKI along with other comorbidities that patients might suffer from, make it really problematic to draw a conclusion. Also, animal models used in the experimental studies are more or less comparable to the human body from the physiological and pathophysiological point of view. Recent data shows that OS plays a crucial role in both initiating and further development of AKI, making it a possible target for therapies. Oxidative damage leading to DNA and RNA alterations, peroxidation of lipids, changes in the structure of proteins, represents an important cause of kidney damage that must be addressed when attempting to treat AKI. ROS, RNS, FRs are closely linked to AKI, and they must be very well understood in order to be targeted. Endogenous antioxidants represent a first line against oxidative damage, and their downregulation worsens the outcome in kidney injury. Antioxidant supplementation is intensely studied, showing promising renoprotective effects. Moreover, research focuses on OS biomarkers that can be detected in early phases of AKI, with apparently good results, whether we talk about detecting novel molecules or using newly-constructed monoclonal antibodies. Nonetheless, more studies are needed in order to establish new clinical guidelines.

Oxidative stress in AKI, although accepted as a key component in the pathophysiology of AKI, is yet to be understood as a target for novel therapeutics.

### Funding
The authors received no funding for this work.

### Competing Interests
The authors declare that they have no competing interests.

### Author Contributions
- Anamaria Magdalena Tomsa conceived and designed the experiments, prepared figures and/or tables, approved the final draft.
- Alexandru Leonard Alexa conceived and designed the experiments, prepared figures and/or tables, approved the final draft.
- Monica Lia Junie analyzed the data, authored or reviewed drafts of the paper, approved the final draft.
- Andreea Liana Rachisan analyzed the data, authored or reviewed drafts of the paper, approved the final draft, she participated in the correction of the whole manuscript.
- Lorena Ciumarnean analyzed the data, authored or reviewed drafts of the paper, approved the final draft.

### Data Availability
This is a review article.

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
