# Peer review of "Oxidative stress as a potential target in acute kidney injury"

_PeerJ, doi:10.7717/peerj.8046_

## Round 0.1 · original submission · Major Revisions

As noticed by Reviewer 2, your manuscript appears as duplication of one previous publication. I perfectly agree with the Reviewer that you need to differentiate it.

Please change your manuscript according to all the comments before re-submitted it.

Reviewer 1 ·

Basic reporting

no comment

Experimental design

no comment

Validity of the findings

no comment

Additional comments

This is a timely and relevant review addressing the role of oxidative stress in acute kidney injury.
Some modifications can be done to improve the manuscript:
-When describing the physiological role of NO, more studies should be discussed, including those showing the role of NO in tubuloglomerular feedback and renal blood flow autoregulation.
-The review may benefit from a new section in which the authors could discuss recent studies regarding novel experimental strategies targeting oxidative stress in models of AKI and the state of art in the identification of new pharmacological targets in oxidative stress area.
-The relevance of references 43 and 47 in oxidative stress mediated injury during AKI needs to be further detailed.

- The “Sepsis induced AKI” section needs to be improved; it lacks information on how oxidative stress is relevant to this type of AKI.

-In the ischemia/reperfusion injury section, a conclusion paragraph needs to be added, so that the reader has a general idea on what sources of ROS are responsible for oxidative stress during AKI and which mechanisms are turned on to try to reduce oxidative damage.

-Furthermore, I suggest the addition of a Figure in which the authors summarize the sources of ROS in AKI, anti-oxidant mechanisms and possible targets.

-Apoptosis is misspelled in the abstract.

Reviewer 2 ·

Basic reporting

This is a review of oxidative stress in acute kidney injury. However, the title is almost the same published review article ("Role of Oxidative Stress in Drug-Induced Kidney Injury." Int J Mol Sci, 17: E1826, 2016). Then, it is better to change the title such as "The involvement of oxidative stress in acute kidney injury: an overview." The authors should avoid the similar title with published article.

1.The authors should add the ilustration about antioxidant in the kidney (lines 266-337).

2. As for oxdative stress biomarker (lines 339-350), descritpion should be expanded. For example, oxidative renal biomarker, urinary vanin-1, has been reported in animal models with acute kidney injury (J Pharmacol Exp Ther. 2012 Jun;341(3):656-62.).

Experimental design

N/A

Validity of the findings

N/A

Additional comments

This is a review of oxidative stress in acute kidney injury. However, the title is almost the same published review article ("Role of Oxidative Stress in Drug-Induced Kidney Injury." Int J Mol Sci, 17: E1826, 2016). Then, it is better to change the title such as "The involvement of oxidative stress in acute kidney injury: an overview." The authors should avoid the similar title with published article.

1.The authors should add the ilustration about antioxidant in the kidney (lines 266-337).

2. As for oxdative stress biomarker (lines 339-350), descritpion should be expanded. For example, oxidative renal biomarker, urinary vanin-1, has been reported in animal models with acute kidney injury (J Pharmacol Exp Ther. 2012 Jun;341(3):656-62.).

Reviewer 3 ·

Basic reporting

Authors tried to review the oxidative stress in AKI. I found this article is difficult to read and many contents has already included in reference #21 (
Oxidant Mechanisms in Renal Injury and Disease, Ratliff BB, Abdulmahdi W, Pawar R, Wolin MS. Antioxid Redox Signal. 2016 Jul 20;25(3):119-46).

My recommendation is “Major revision” to improve the followings;
1. English language.
2. Text arrangement.
3. Full name should appear the first time in the text along with abbreviations.
4. Many references cited are outdated and many are review articles.
5. Sessions 3 and 4 are very short, only one paragraph each.
6. Need tables and figures to enhance readership.

I suggest change the title to only cover recent advances (in the past 5-years period) in this field as reference #21 (dated 2016) already have a through and comprehensive coverage.

Experimental design

N/A

Validity of the findings

N/A

Additional comments

My recommendation is “Major revision” to improve the followings;
1. English language.
2. Text arrangement.
3. Full name should appear the first time in the text along with abbreviations.
4. Many references cited are outdated and many are review articles.
5. Sessions 3 and 4 are very short, only one paragraph each.
6. Need tables and figures to enhance readership.

I suggest change the title to only cover recent advances (in the past 5-years period) in this field as reference #21 (dated 2016) already have a through and comprehensive coverage.

---

## Round 0.2 · accepted · Accept

The manuscript has been evaluated by the Reviewer 1 who considered it suitable for publication.

Reviewer 1 ·

Basic reporting

The authors addressed all my comments.

Minor:
Table 1. Please check sources of nitric oxide. eNOS is mentioned twice.

Experimental design

No comment

Validity of the findings

No comment

Additional comments

The authors addressed all my comments.

Minor:
Table 1. Please check sources of nitric oxide. eNOS is mentioned twice.